# Study on Penetration Performance of Rear Shaped Charge Warhead

**DOI:** 10.3390/ma14216526

**Published:** 2021-10-29

**Authors:** Yanan Du, Guanglin He, Yukuan Liu, Zhaoxuan Guo, Zenghui Qiao

**Affiliations:** 1Science and Technology on Electromechanical Dynamic Control Laboratory, Beijing Institute of Technology, Beijing 100081, China; 3120205105@bit.edu.cn (Y.D.); yukuanliu@bit.edu.cn (Y.L.); 3120190185@bit.edu.cn (Z.G.); 2Strategic Support Force, Beijing 100081, China; yanxiaofang1985@gmail.com

**Keywords:** shaped charge jet, simulation cabin, numerical simulation, experimental verification, penetration performance

## Abstract

In guided munitions, the shaped charge jet (SCJ) warhead is located behind the simulation compartment (including the control cabin, the steering gear cabin, and the guidance cabin). Therefore, the order of penetration of the SCJ is the simulation cabin and the target. To study the penetration performance of the SCJ to the target plate, the numerical simulation method is used to study the penetration performance of the designed warhead for the steel target at different standoffs, and the depth of penetration (DOP) at the best standoff is obtained, that is, the DOP of the steel target is about 128 mm. Additionally, the penetration performance of the SCJ warhead to target is studied by numerical simulation and experimental verification. Numerical simulation and experimental results show that the DOP of the SCJ warhead to the steel target is 50 mm without the simulation cabin, and about 30 mm with the simulation cabin. The results show that the penetration performance of SCJ is greatly weakened under the condition of non-optimal standoff, but the rear shaped charge warhead still has a strong penetration performance after completing the penetration of the simulated cabin.

## 1. Introduction

High-explosive anti-tank (HEAT) munitions use the shaped charge effect of hollow charge to crush the liner and form a high-speed metal jet to break through armor. They are widely used in all kinds of ammunition against armor targets [1]; the depth of penetration (DOP) is widely studied as the core content of the armor-piercing power of shaped charge warhead. Many factors affect the DOP of the shaped charge jet (SCJ) warhead. For example, the parameters of the liner, the parameters of the charge, the standoff, the detonation point and the target type, the angle of the penetration target, the intervention of external conditions, etc., all have varying degrees of impact on the DOP of the SCJ warhead.

The standoff has an important influence on the SCJ penetration into the target plate. If the standoff is too small, the jet is not completely formed, and the penetration ability is limited. If the standoff is too large, the contradiction between jet elongation and fracture will appear [2]. On the one hand, with the increase in the standoff, the jet elongates, so as to improve the penetration depth. On the other hand, with the increase in the standoff, the jet produces radial dispersion and swing, and fracture occurs when it extends to a certain extent, and the penetration depth decreases [3,4]. Therefore, many scholars at home and abroad have done a lot of research on the impact of the standoff. Zhu et al. [5] studied the influence of the standoff on the penetration of the SCJ with a cylinder cone combined cover through experiments and numerical simulation and obtained the favorable standoff of the SCJ with a cylinder cone combined cover. Held [6,7] used the experimental method to analyze the penetration model of arbitrary charge to the target plate under the conditions of a continuous jet formed at a close distance, a continuous partial granulation jet formed at medium standoff, and complete granulation jet formed at large standoff. The research results show that for a specific target plate, a certain shaped charge has an optimal standoff corresponding to the optimal penetration depth. In addition, the penetration performance of shaped charge jet can also be affected by internal factors and external intervention factors. Literature [8,9,10,11] studied the influence of the electromagnetic effect on the penetration performance of the SCJ. By controlling the direction of the electromagnetic field, the penetration ability of the SCJ can be improved or reduced. Pyka et al. [12], through theoretical research and numerical tests on two types of the Polish ŁK cumulative charge, showed that the jet still retains its ability to penetrate after penetrating the shield. It follows that this ŁK charge can effectively pierce shields of this type with a thickness of more than three calibers. It has been observed that peripherally initiated charges have a positive effect on the jet growth in the procession of the collapsing liner. Zochowski et al. [13], through theoretical research and numerical experiments, showed that the penetration capability of warheads and the materials of explosives, targets, and warheads will affect the penetration performance of the warhead. At the same time, the proposed numerical simulation method improves the definition of the numerical model of warhead based on a real projectile, which is helpful to accurately reproduce the formation and target penetration process of a shaped charge jet. Shvetsov et al. [8] have shown through theoretical research and calculations that SCJ can significantly reduce the penetration depth of the focused jet on the target plate under the action of a magnetic field. Ma et al. [9,10,11] analyzed the acceleration mechanism of SCJ under the action of an electromagnetic field by establishing a theoretical model and verified through experiments that the application of a magnetic field in the inertial stretching stage of SCJ can improve the stability of the jet, thereby improving the DOP of the jet. The characteristics of the SCJ warhead itself have a great influence on the DOP of the target plate, but different types of target plates also have a great influence on the penetration performance of the SCJ warhead. Jia et al. [14,15] studied the penetration performance of the SCJ warhead on woven fabric rubber composite armor through theoretical analysis and a DOP experiment. The research shows that Kevlar woven fabric can effectively enhance the anti-penetration ability of rubber composite armor. Wu et al. [16] studied the penetration performance of the SCJ warhead into ultra-high-performance concrete (UHPC), 45#steel, and their composite targets through experimental verification and numerical simulation. The results show that the anti-penetration ability of the UHPC/45 steel composite target is better than that of the UHPC and 45#steel target, and the anti-penetration ability of spaced target is better than that of the stacked target. 

It can be seen from the above references that researchers have done much research on the penetration performance of the SCJ warhead. Literature [2,3,4,5,6,7] studied the influence of blasting height on the penetration depth of shaped charge jet, [8,9,10,11] studied the influence of external factors on the penetration performance of shaped charge jet, and [12,13,14,15] studied the influence of target material and structure on the penetration ability of the SCJ. From the research conclusions of [16] and [17], it can be seen that the spaced target has a stronger ability to resist the penetration of the SCJ than the laminated target, because the SCJ warhead designed in this research was placed behind the guided munitions; therefore, there are the following two problems. First, the total length of the simulated cabin is about 660 mm, whereas the diameter of the designed liner is 65 mm (LD = 65 mm) after the SCJ has penetrated the simulated cabin section, the distance from the bottom diameter of the liner to the target plate is about 10 LD. Second, after the SCJ has penetrated the simulated cabin (includes control cabin, steering gear cabin, and guidance cabin), it can begin to penetrate the target. Because the components in the simulated cabin are equivalent to spaced targets, the jet will dissipate more energy during the penetration process, thereby reducing the DOP of the SCJ on the target plate. Therefore, to study the penetration performance of the rear SCJ warhead, first, the DOP at different standoffs was studied through the method of numerical simulation, Second, the method of combining numerical simulation and experimental verification is used to study the DOP of the SCJ warhead on the target plate when the simulated cabin is not installed and when the simulated cabin is installed. Finally, the DOP at the best standoff, conditions when the simulation cabin is not installed and then is installed are compared and analyzed.

## 2. Jet Penetration Theory 

### 2.1. Theory of the SCJ Breaking Armor 

The DOP is the core content of penetration power of the SCJ warhead. Since the emergence of the SCJ warhead, the theory of DOP calculation has been developed for more than 60 years, and a series of theoretical models have been established [1], that is, the steady ideal incompressible hydrodynamics theory, the quasi steady ideal incompressible hydrodynamics theory, the incompressible hydrodynamics theory considering the strength of target plate, and the hydrodynamics theory of broken jet penetration.

The armor breaking formula under the steady ideal incompressible hydrodynamics theory is shown in Equation (1).
(1)$$L=l\sqrt{\frac{\rho_{j}}{\rho_{t}}}$$
where *L* is the penetration depth, *l* is the jet length, $\rho_{j}$ is the jet density, and $\rho_{t}$ is the target density.

The armor breaking formula under the quasi steady ideal incompressible hydrodynamics theory is shown in Equation (2).
(2)$$L=\left(H-b\right)\left[{\left(\frac{v_{j0}}{v_{jp}}\right)}^{\sqrt{\frac{\rho_{j}}{\rho_{t}}}}-1\right]$$
where *L* is the penetration depth, *H* is the standoff, (H−b) is the fixed value, $v_{j0}\;$ is the velocity of the jet head, $v_{jp}$ is the tail velocity of the jet,$\;\;\rho_{j}\;\;$ is the jet density, and $\rho_{t}$ is the target density.

The armor breaking formula under the incompressible hydrodynamics theory considering the strength of target plate is shown in Equation (3).
(3)$$L=\left(t_{0}-t_{A}\right)v_{j}\frac{T_{0}}{T}{\left[\frac{T+\sqrt{T^{2}-{\left(1-c\right)}^{2}v_{jc}^{2}}}{T_{0}+\sqrt{T_{0}-{\left(1-c\right)}^{2}v_{jc}^{2}}}\right]}^{\frac{-1}{\sqrt{c}}}-H+b$$
where $T_{0}$ = $-cv_{j0}+\sqrt{cv_{j0}^{2}+\left(1-c\right)v_{jc}^{2}}$, *T* = $-cv_{j}+\sqrt{cv_{j}^{2}+\left(1-c\right)v_{jc}^{2}}$, *c* = $\sqrt{\frac{\rho_{t}}{\rho_{j}}}$, $t_{0}$ indicates the start time of jet armor breaking, $t_{A}$ is the jet generation time, $\rho_{j}\;$ is the jet density, $\rho_{t}$ is the target density, $v_{j0}$ is the velocity of the jet head, $v_{jp}$ is the tail velocity of the jet, $v_{jc}$ is critical velocity of jet, *L* is the penetration depth, *H* is the standoff, and (H−b) is the fixed value.

The armor breaking formula under the hydrodynamics theory of broken jet penetration is shown in Equation (4).
(4)$$L=\frac{1}{Kc}\left[\sqrt{cv_{j0}^{2}+\left(1-c\right)v_{jc}^{2}}-\sqrt{cv_{j}^{2}+\left(1-c\right)v_{jc}^{2}}\right]-\frac{v_{jc}}{K\sqrt{c}}\left\{\arctan{\left[\frac{cv_{j0}^{2}+\left(1-c\right)v_{jc}^{2}}{cv_{jc}^{2}}\right]}^{\frac{1}{2}}-\arctan{\left[\frac{cv_{j}^{2}+\left(1-c\right)v_{jc}^{2}}{cv_{jc}^{2}}\right]}^{\frac{1}{2}}+\arctan\left(\frac{\sqrt{c}v_{j0}}{v_{jc}}\right)-\arctan\left(\frac{\sqrt{c}v_{j}}{v_{jc}}\right)\right\},$$
where *L* is the penetration depth, *K* is the velocity gradient, *K* = $\frac{dv_{j}}{dl}$, *c* = $\sqrt{\frac{\rho_{t}}{\rho_{j}}}$, $\rho_{j}$ is the jet density, $\rho_{t}$ is the target density, $v_{j0}$ is the velocity of the jet head, $v_{jp}$ is the tail velocity of the jet, $v_{jc}$ is critical velocity of jet, and *l* is the jet length.

Among the four penetration theories, the results are consistent with the experimental results at the initial stage of jet penetration. With the increase in the DOP, the fourth penetration theory is closer to reality. The core of the fourth penetration theory is that the jet does not elongate after fracture, the velocity gradient of the jet remains unchanged since fracture, and the jet after fracture becomes unstable and overturns in the process of motion, which has a certain impact on the penetration of the jet.

### 2.2. The Jet Penetrates the Multi-Layer Target 

The research on the penetration performance of the SCJ on multilayer targets has not yet seen an authoritative theoretical formula. At present, numerical simulation and experimental comparison methods are mainly used. For example, the DOP of the SCJ into spaced targets and stacked targets with the same thickness is studied by numerical simulation and experiment. Relevant studies have shown that the spaced target’s ability to resist the penetration of shaped jets is stronger than that of stacked targets [16].

The depth of jet penetration into the multi-layer target is expressed by Equation (5).
(5)$$H=\Phi f+H_{L}$$
where *H* is the total penetration depth, Φ is the thickness of target plate, *f* is the number of jets penetrating the target plate, specifically the number of layers penetrating the target plate, and *H*_L_ is the penetration depth of the last layer target plate.

## 3. Structural Model 

### 3.1. Warhead Structure Design 

The charge diameter of the SCJ warhead designed in this research is 75 mm and the charge height is 100 mm. To improve the jet length and the DOP as much as possible, the liner structure is designed with variable wall thickness and cone angle. Variable wall thickness means that the thickness of the liner increases gradually from the top to the bottom. The thickness at the cone angle of the top of the liner is 1.4 mm, and the thickness at the bottom of the liner is 2 mm. Variable cone angle means that the angle of liner increases gradually from top to bottom: the cone angle from top to middle of liner is 38°, and the cone angle from the middle to the bottom of the liner is 60°. The designed shaped charge warhead is shown in Figure 1.

### 3.2. Penetration Model 

Because the warhead of the shaped jet is located behind the simulated cabin, the order of penetration of the SCJ is simulated cabin and target plate. The length of the simulated cabin is 660 mm, and the diameter of the liner is 65 mm. Therefore, The SCJ can only penetrate the target plate after completing the 660 mm simulated cabin, that is to say, the penetration of the SCJ into the target plate is completed at 10 LD of the standoff, whereas the optimal standoff is 1–3 LD. Therefore, to study the DOP of the rear SCJ warhead into the target plate at the optimal standoff, the established penetration model is shown in Figure 2.

In Figure 2, five different standoffs are designed: 1 LD, 1.5 LD, 2 LD, 2.5 LD, and 3 LD. The target plate is 150 mm Q235 steel.

The penetration model without the simulation cabin is shown in Figure 3 when the standoff is 10 LD.

In Figure 3, because the distance between warhead and target exceeds the optimal standoff, the penetration ability of the SCJ begins to weaken. To better observe the penetration performance of the SCJ, the target plate is set as three layers of Q235 steel plate with different thicknesses, in which the thickness of the first layer of steel target is 50 mm, the thickness of the second layer of steel target is 20 mm, and the thickness of the third layer of steel target is 10 mm.

The penetration model with the simulation cabin is shown in Figure 4 when the standoff is 10 LD.

In the structural model of the SCJ warhead penetration simulation cabin shown in Figure 4, the simulation cabin includes the control cabin (1–5 layers), the steering gear cabin (6–18 layers), and the guidance cabin (19–25 layers). The number, name, thickness of each layer of slices, and the distance between each layer in the simulated cabin are shown in Table 1. The distance between the liner and the first target plate is 30 mm.

## 4. Numerical Simulation

When studying the problem of contact explosion interaction, the Lagrange algorithm can be used to define the contact between the explosive and the structure to consider the interaction; however, in the Lagrange algorithm, the grid and the material are interconnected. When the material flows, the shape of the mesh will also change. When a large deformation occurs, the material flow is significant, causing the mesh nodes to follow the material flow and produce a large displacement, resulting in serious distortion of the mesh. Therefore, the explosive element is prone to serious distortion in the process of explosion, resulting in the interruption of calculation. When the FSI (fluid-structure interaction) method is used to calculate the effect of the explosive on the structure, the Euler algorithm is used for explosive, whereas the Lagrange algorithm is used for structure, which can avoid the problem of mesh distortion. In this study, the numerical model is composed of explosive, liner, air, simulation cabin, and target plate. The explosive, liner, and air are modeled by Eulerian grid (fluid structure), the element is modeled by multi material group ALE (Arbitrary Lagrange-Euler) algorithm, and the simulation cabin and target plate are modeled by Lagrangian grid (solid structure); the FSI algorithm is used between fluid structure and solid structure material. 

### 4.1. Finite Element Model

The penetration model established in Section 3.2 was imported into HyperMesh (version 2017) for meshing. The liner mesh, charge mesh, and air domain mesh are mainly composed of hexahedral mesh and prismatic pentahedral mesh. The interface between the liner mesh and charge mesh adopts common node, and the interface between the liner mesh and air domain adopts common node, the interface between charge grid and air domain grid adopts common node. The simulation cabin and steel target use hexahedral mesh. Because the jet diameter is small, the penetration area is mainly concentrated in the center of the target plate. Therefore, in the radial direction of the target plate, the mesh size gradually increases from the center to the edge. In the axial direction of the target plate, the mesh size is fixed at1 mm. Because the numerical simulation is based on the FSI algorithm, there must be overlap between the fluid grid and the solid grid. Because the penetration model is a centrosymmetric structure, to reduce the amount of calculation, the quarter finite element model is established as shown in the Figure 5.

Figure 5a represents the finite element model with 530,250 meshes under the standoff, which is 1–3 LD; Figure 5b represents the finite element model without the simulation cabin, with 605,500 meshes; and Figure 5c represents the finite element model with the simulation cabin, with 800,970 meshes.

### 4.2. Material Constitutive Model and Parameters

In the finite-element model as shown in Figure 3, 8701 explosive is selected [18], high explosive material model and the JWL (Jones Wilkins Lee) state equation are used, material parameters are shown in Table 2, and the JWL state equation expression is shown in Equation (6).

General pressure expression of JWL equation of state [19].
(6)$$p=A\left(1-\frac{\omega }{r_{1}V}\right)e^{-r_{1}V}+B\left(1-\frac{\omega }{r_{2}V}\right)e^{-r_{2}V}+\frac{\omega E}{V}$$
where *A*, *B*, $r_{1}$, $r_{2}$, and ω are material constants; *V* is the initial relative volume; *E* is the initial specific internal energy; ρ is the initial explosive density; $P_{CJ}$ is the detonation pressure; D is the detonation speed; and p is the hydrostatic pressure.

In this study, the liner material is copper, the material model is JOHNSON_COOK, and the state equation is described by GRUNEISEN. The material parameters are shown in Table 3, the expression of the GRUNEISEN equation of state in compression state is shown in Equation (7), and the expression in expansion state is shown in Equation (8).

Expression of GRUNEISEN equation of state in the compressed state [20,21]:(7)$$p=\frac{\rho_{0}C^{2}\mu \left[1+\left(1-\frac{\gamma_{0}}{2}\right)\mu -\frac{\alpha }{2}\mu^{2}\right]}{{\left[1-\left(S_{1}-1\right)\mu -S_{2}\frac{\mu^{2}}{\mu +1}-S_{3}\frac{\mu^{3}}{{\left(\mu +1\right)}^{2}}\right]}^{2}}+\left(\gamma_{0}+\alpha \mu \right)E$$

Expression of GRUNEISEN equation of state in expansion state:(8)$$p=\rho_{0}C^{2}\mu +\left(\gamma_{0}+\alpha \mu \right)E$$
where *E* is the initial internal energy, *C* is the intercept of the $v_{s}-v_{p}$ curve, $S_{1}$,$\;S_{2}$, and $S_{3}$ are the coefficients of the slope of the $v_{s}-v_{p}$ curve, $\gamma_{0}\;$ is the GRUNEISEN coefficient, and α  is the first-order volume, correction of $\gamma_{0}$.

The material model in the simulation cabin adopts PLASTIC_KINEMATIC, and the material parameters are shown in Table 4. The model is suitable for isotropic and kinematic hardening plastic models, optionally including rate effects. This is a very cost-effective model.

MAT_NULL model is adopted for air; air material parameters are shown in Table 5. The equation of state is described by linear polynomials, which is EOS_LINEAR_ POLYNOMIAL. This material allows equations of state to be considered without computing deviatoric stresses. Optionally, a viscosity can be defined. In addition, erosion in tension and compression is possible.

### 4.3. Numerical Simulation Results

The K file generated by the finite element model is submitted to LS-DYNA (version 19.0) for solution, and the penetration results of the SCJ warhead at 1–3 LD of the standoff are obtained, as shown in Figure 6.

Figure 6a shows the numerical simulation results where the standoff is 1 LD, and the DOP of the SCJ into Q235 steel target is about 62 mm; Figure 6b shows the numerical simulation results where the standoff is 1.5 LD, and the DOP of the SCJ into Q235 steel target is about 82 mm; Figure 6c shows the numerical simulation results where the standoff is 2 LD, and the DOP of the SCJ into Q235 steel target is about 128 mm; Figure 6d shows the numerical simulation results where the standoff is 2.5 LD, and the DOP of the SCJ into Q235 steel target is about 122 mm; and Figure 6e shows the numerical simulation results where the standoff is 3 LD, and the DOP of the SCJ into Q235 steel target is about 109 mm.

It can be seen from Figure 6 that when the standoff is between 1–3LD, the DOP of the SCJ into the steel target increases first and then decreases; that is, when the standoff increases from 1 LD to 1.5 LD, the DOP increases from 62 mm to 82 mm, an increase of 32.3%; when the standoff increases from 1.5 LD to 2 LD, the DOP increases from 82 mm to 128 mm, an increase of 56.1%; when the standoff increases from 2 LD to 2.5 LD, the DOP decreases from 128 mm to 122 mm, a decrease of 4.7%; when the standoff increases from 2.5 LD to 3 LD, the DOP decreases from 122 mm to 109 mm, a decrease of 10.7%. When the standoff is 2 LD, the DOP reaches the maximum value of 128 mm. Therefore, the best standoff of the SCJ warhead designed in this study is 2 LD.

The penetration results without the simulation cabin are shown in Figure 7 when the standoff is 10 LD.

Figure 7a represents the shape of the SCJ at t = 83.2 μs; at this time, the jet is intact and there is no necking or fracture. Figure 7b represents the shape of the SCJ at t = 94.4 μs, after the SCJ extends to a certain extent in the air, the jet begins to appear necking and fracture; in other words, from this moment, the SCJ will gradually break into small segments. Figure 7c represents the shape of the SCJ at t = 94.4 μs; the jet at this moment has been broken into multiple small segments, and the length of the jet after the fracture does not change. Therefore, the distance between each segment of the jet increases gradually, and the jet at this time begins to penetrate the first layer of 50 mm thick steel plate. Figure 7d represents the shape of the SCJ at t = 132.8 μs; because the fractured jet penetrates the target plate at intervals, the stress state of the previous jet disappears, and the additional energy is consumed in the subsequent jet penetration, which leads to the insufficient penetration energy of the residual jet after the first layer of target is penetrated. As shown in Figure 7e, the size of the inlet hole of the first layer target plate is 54.3 mm × 46.5 mm, and the size of the outlet hole is 36.1 mm × 14.7 mm; after the SCJ has penetrated the first layer of target plate, the residual jet does not continue to penetrate the second target plate, but accumulates on the second target plate, leaving pits on the second target plate.

Therefore, the effective DOP of the SCJ warhead without the simulation cabin is 50 mm. Compared with the DOP at the optimal standoff, the DOP at the 10 LD standoff is reduced by approximately 61%. According to the fourth penetration theory, due to the increase in the standoff, the length of the jet extends to a certain extent and then necks and breaks; the jet is no longer continuous and overturns when it moves in the air. Therefore, when the broken jet penetrates into the target plate, it needs to “open a hole” again, which will consume more energy, resulting in the decrease in the DOP to the steel target. That is to say, when the SCJ warhead designed in this study is not at the optimal standoff, the impact on the penetration ability of steel target is 61%.

The penetration results with the simulation cabin are shown in Figure 8 when the standoff is 10 LD.

In Figure 8a indicates that t = 20.8 μs, and the SCJ has not yet begun to penetrate the target plate; Figure 8b indicates that t = 36.8 μs, when the SCJ has completed its penetration into the control cabin, and the SCJ is intact at this time; Figure 8c indicates that t = 112 μs, at this time the SCJ completes its penetration into the rudder engine compartment, and the jet is intact; Figure 8d indicates t = 112.8 μs, at this time the jet has completed its penetration into the steering gear, and the SCJ begins to show necking and fracture; Figure 8e indicates that t = 164.8 μs, at this time the SCJ completes its penetration into the guidance cabin, and the SCJ breaks; Figure 8f indicates that t = 180.8 μs, when the SCJ completes its penetration into the 50 mm steel plate, and the DOP is about 30 mm, and the residual jet energy can no longer penetrate the steel target, that is, after the SCJ completes the penetration of the simulated cabin, the maximum DOP of the residual jet is about 30 mm.

Compared with the DOP when the simulated cabin is not installed, the DOP of the SCJ warhead on the steel target is reduced by about 40%; in other words, the negative impact of the simulated cabin on the penetration capability of the SCJ warhead is about 40%. Compared with the DOP under the best standoff situation, the DOP of the SCJ warhead on the steel target is reduced by 76.5%. In other words, the penetration capability of the SCJ warhead designed by this research on the Q235 steel target is reduced by 76.5% under the combined influence of the standoff and the simulated cabin. 

## 5. Experimental Verification

From the above numerical simulation results, it can be seen that the best standoff of the SCJ warhead designed in this research is 2 LD; however, because the SCJ warhead designed in this study is placed behind the simulation cabin, and the length of the simulation cabin is about 660 mm, which is 10 LD, the SCJ warhead can start to penetrate Q235 steel target only after it has penetrated the simulated cabin, that is to say, the SCJ warhead can penetrate Q235 steel target at 10 LD. The numerical simulation shows that the DOP of the SCJ into Q235 steel target is about 50 mm when the simulation cabin is not installed, and the DOP of the SCJ into Q235 steel target is about 30 mm when the simulation cabin is installed. To verify the DOP of the Q235 steel target by the SCJ warhead when the simulated cabin is not installed and the simulated cabin is installed at the 10 LD standoff, the following two experiments were carried out.

### 5.1. Direct Penetration Test

#### 5.1.1. Test Setup

Penetration target experiment setup without the simulation cabin is shown in Figure 9, with the standoff at 10 LD.

Figure 9a shows the experimental principle diagram, and Figure 9b shows the experimental setup. To ensure the penetration effect of SCJ on the target, we tamped the test site, placed a layer of test device, and measured the flatness with a level meter. From the bottom to the top, the test equipment consisted of a steel plate, test bench, steel plate, 10 mm Q235 steel target, 20 mm Q235 steel target, 50 mm Q235 steel target, wooden stool for the warhead, and the warhead.

#### 5.1.2. Test Result

After the experiment was ready, the warhead was detonated, and the Q235 steel target was recovered after the experiment. Vernier calipers were used to measure the inlet diameter and outlet diameter of the SCJ penetrating the first Q235 steel target. The measurement results are shown in Figure 10.

Figure 10a represents the size of the jet penetrating the Q235 steel target into the hole, and Figure 10b represents the size of the jet penetrating the Q235 steel target out the hole. It can be seen from Figure 10 that the SCJ can penetrate the first layer of steel target with a thickness of 50 mm; however, the size of the jet inlet and outlet is irregular, as shown in Figure 10a, the inlet size is 54 mm × 46 mm and the outlet size is 35.5 mm × 14.5 mm. According to the fourth SCJ penetration theory, because the distance between the target and the liner is 10 LD, the jet has necked and fractured before it started to penetrate the Q235 steel target. The fracture jet overturns in the movement, which led to the formation of irregular jet inlet and outlet holes after the jet penetrated the Q235 steel target.

After the SCJ penetrated the first Q235 steel target, the penetration effect of the second Q235 steel target is shown in Figure 11.

It can be seen from Figure 11 that the residual jet accumulated in the second Q235 steel target and left penetration traces, but it did not penetrate the second Q235 steel target. This is because the jet velocity after fracture did not increase any more, and the fractured jet consumed more energy when penetrating the first Q235 steel target; therefore, after the jet completed the penetration of the first Q235 steel target, the energy of the residual jet was not enough to continue to penetrate the second Q235 steel target and accumulated on the second Q235 steel target. It can be seen that the maximum DOP of the SCJ warhead was 50 mm when it directly penetrated into the Q235 steel target without the simulated cabin at the 10 LD standoff.

### 5.2. Penetration Simulation Cabin and Target Experiment

#### 5.2.1. Test Setup

Penetration target experiment setup with simulation cabin is shown in Figure 12 when the standoff is 10 LD.

Figure 12a shows the experimental principle diagram, and Figure 12b shows the experimental setup. On the basis of the experiment setup shown in Figure 9, the 50 mm, 20 mm, and 10 mm three-layer Q235 steel targets were replaced with a layer of 50 mm thick Q235 steel targets. We placed the simulation cabin on a 50 mm thick Q235 steel target, as shown in Figure 12c. The simulated cabin is composed of a shell and equivalent target plate of internal parts.

#### 5.2.2. Test Result

After the experiment was ready and the warhead was detonated, simulated cabin debris was collected after the SCJ penetrated the simulated cabin and the Q235 steel target. This debris is shown in Figure 13.

It can be seen from Figure 13 that under the detonation effect of explosives and the penetration of the SCJ, the shell of the simulated cabin exploded into irregularly shaped fragments, and the equivalent target plate of the internal components of the simulated cabin was penetrated by the SCJ.

Figure 14 shows the results after the penetration experiment was completed: the residual jet penetrated the Q235 steel target after the SCJ has penetrated the simulated cabin.

In Figure 14, by measuring the DOP of the SCJ on Q235 steel target, it can be seen that the DOP of the SCJ on Q235 steel target is 29.5 mm, about 30 mm after the penetration of the simulated cabin.

## 6. Conclusions

To study the penetration ability of the rear SCJ warhead, the penetration ability of the rear SCJ warhead to the Q235 steel target under different standoffs was simulated by numerical simulation method, and the optimal standoff of the SCJ warhead was obtained. The penetration ability of the SCJ into the Q235 steel target without and with simulated cabin was studied. In addition, numerical simulation and experimental verification were used to study the penetration ability of the SCJ into Q235 steel target without or with the simulated cabin. The conclusions are as follows: The optimal standoff of the SCJ warhead is 2 LD, and the DOP of the SCJ to the Q235 steel target is about 128 mm.The DOP of the SCJ into the Q235 steel target is about 50 mm at 10 LD standoff without the simulated cabin, compared with the DOP (128 mm) of the Q235 steel target at the optimal standoff, its penetration ability is reduced by about 61%. In other words, the penetration ability of the SCJ is weakened by 61% under the condition of non-optimal standoff.At 10 LD, the DOP of the residual jet into the Q235 steel target is about 30 mm after the penetration of the SCJ into the simulated cabin; compared with the penetration ability without the simulated cabin, the DOP decreases by 40%. In other words, the penetration ability of the SCJ is weakened by about 40% under the influence of the simulated cabin.Compared with the DOP of the Q235 steel target at the optimal standoff, the DOP of the SCJ to the Q235 steel target decreases by 76.5% under the combined influence of non-optimal standoff and simulated cabin, and the final DOP of the SCJ warhead is about 30 mm.When the diameter of the liner is 10 times the blast height, there is a certain error between the breach size of the first layer target plate obtained by the numerical simulation of direct penetration into the target plate and the test value. The error of the entry hole size is 0.3 mm × 0.5 mm and the error of the exit hole size is 0.6 mm × 0.2 mm. The penetration depth obtained by numerical simulation of penetration simulation cabin and target plate is also different from the experimental value, and the penetration depth error is 0.5 mm. The deviation may be caused by impurities in the fluid during the test, or a small amount of impurities on the target surface caused by the environment during the test, which will have an impact on the process of shaped charge jet penetrating the target. At the same time, the accuracy of the numerical simulation is also a reason. However, the error between the target opening size and the penetration depth is within 5%, which is in good agreement as a whole.If you want to eliminate this error, you can improve the mesh quality of the finite element model as much as possible and adopt more accurate numerical simulation calculations when the computer performance allows.

## Figures and Tables

**Figure 1 materials-14-06526-f001:**
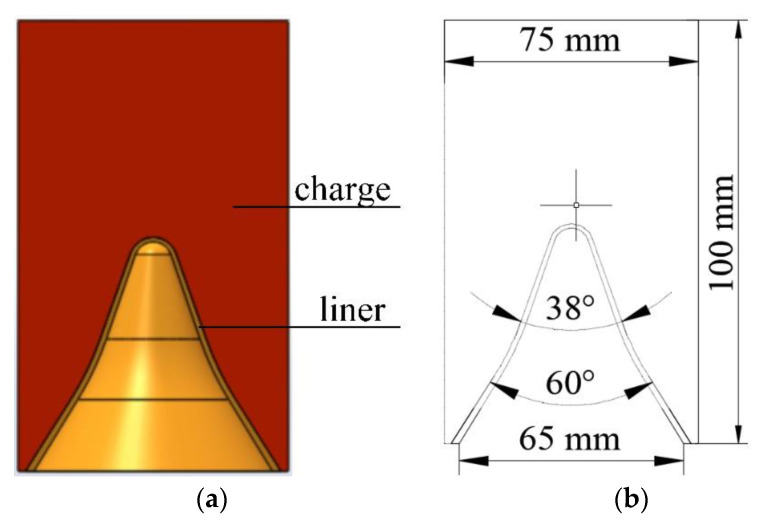
Structure diagram of the SCJ warhead: (**a**) 3D drawing of warhead, and (**b**) structural dimension drawing of warhead.

**Figure 2 materials-14-06526-f002:**
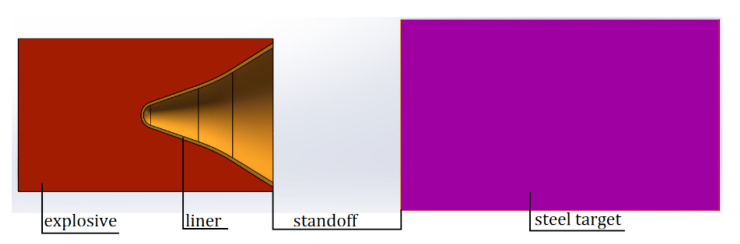
Penetration model of the standoff is 1–3 LD.

**Figure 3 materials-14-06526-f003:**
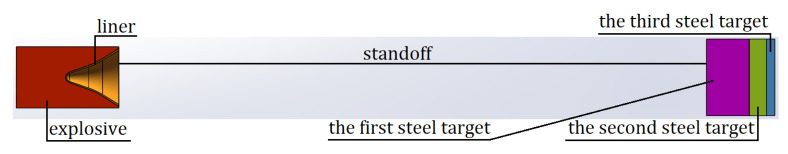
Penetration model without the simulation cabin.

**Figure 4 materials-14-06526-f004:**
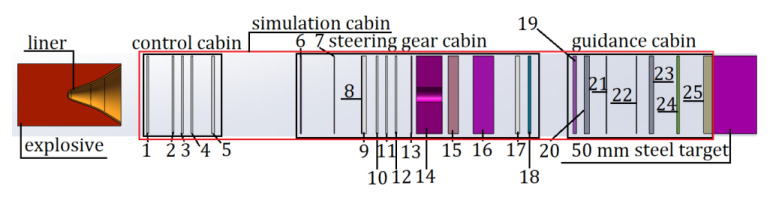
Penetration model with the simulation cabin.

**Figure 5 materials-14-06526-f005:**
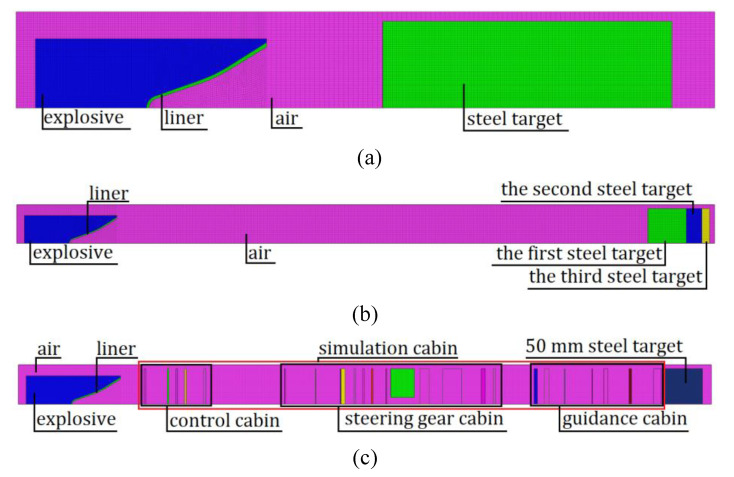
Finite element model: (**a**) of striking plate, (**b**) without simulation cabin, and (**c**) with the simulation cabin.

**Figure 6 materials-14-06526-f006:**
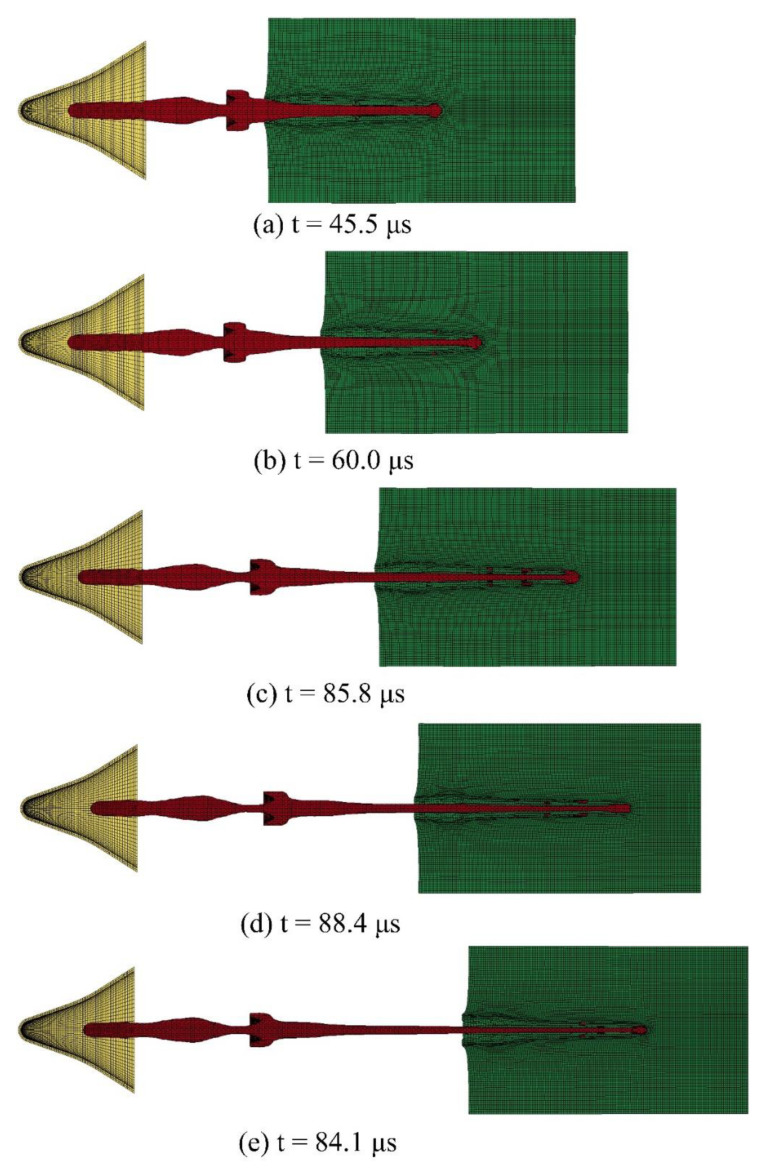
Penetration result of the standoff is 1–3 LD.

**Figure 7 materials-14-06526-f007:**
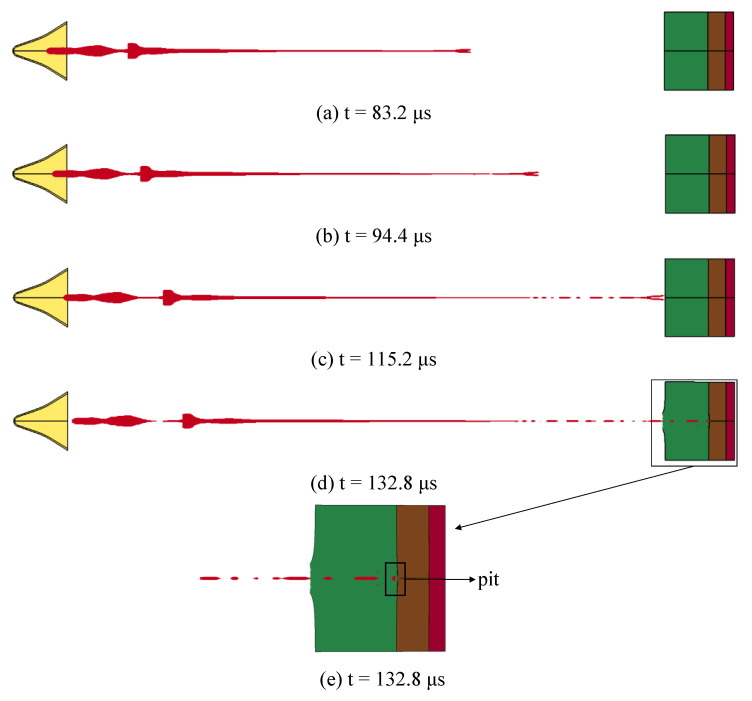
Penetration result without the simulation cabin.

**Figure 8 materials-14-06526-f008:**
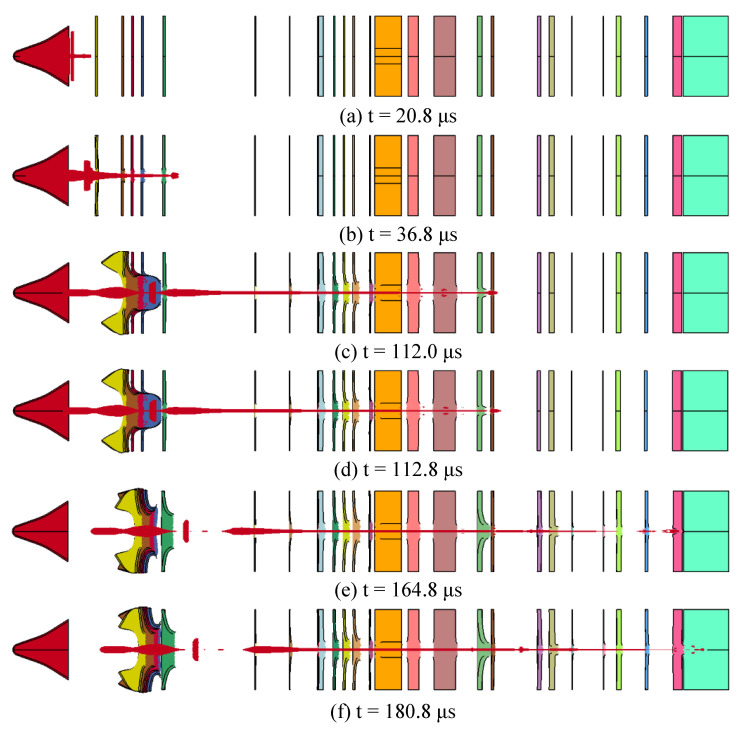
Penetration result with the simulation cabin.

**Figure 9 materials-14-06526-f009:**
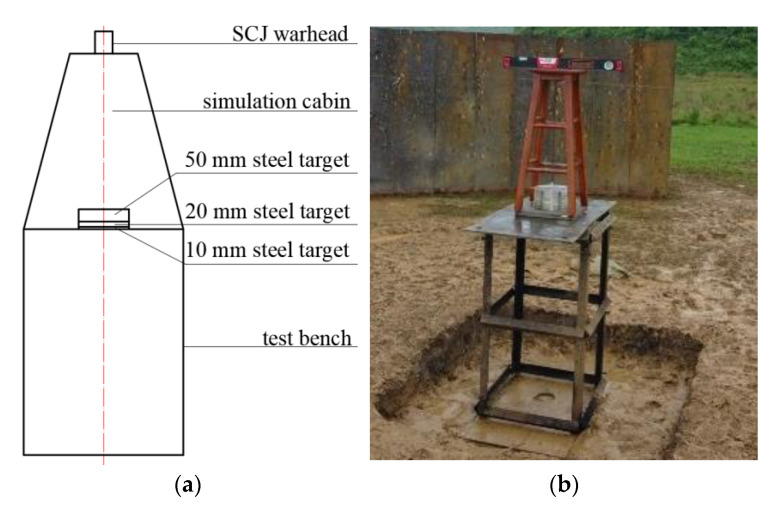
Penetration experiment setup without the simulation cabin: (**a**) schematic diagram of no cabin test setup, (**b**) no cabin test drawing.

**Figure 10 materials-14-06526-f010:**
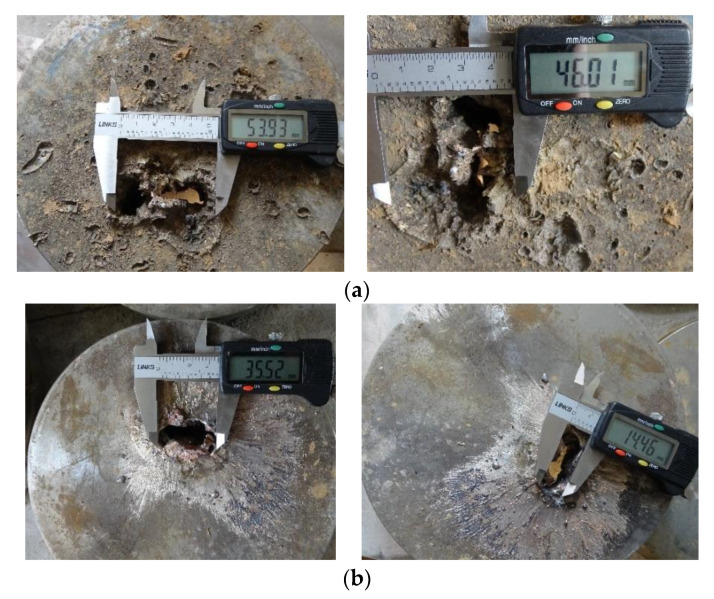
Penetration result of the Q235 steel target without the simulation cabin: (**a**) the size of the jet penetrating the Q235 steel target into the hole, (**b**) the size of the jet penetrating the Q235 steel target into the hole.

**Figure 11 materials-14-06526-f011:**
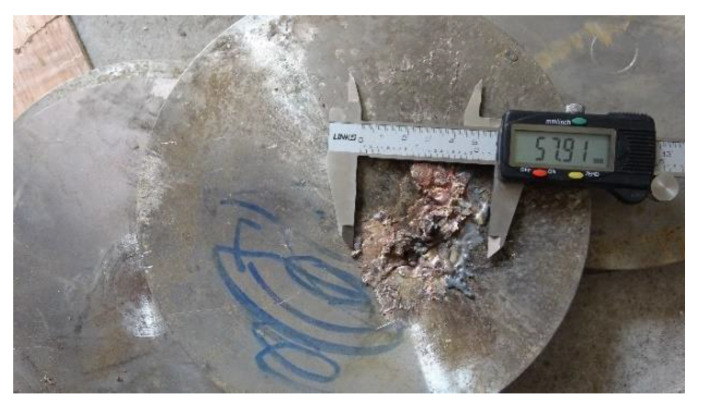
Penetration effect of residual jet.

**Figure 12 materials-14-06526-f012:**
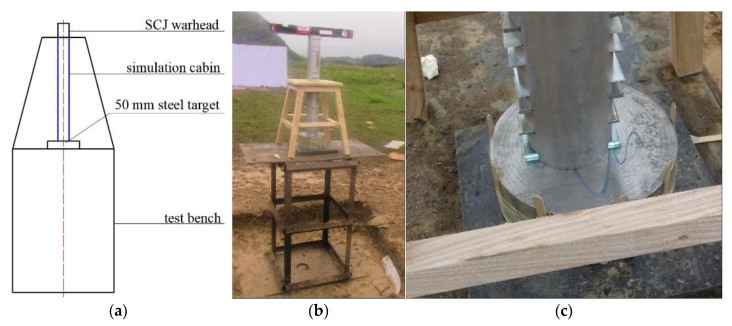
Penetration experiment setup with simulation cabin: (**a**) the experimental principle diagram, (**b**) the experimental setup, and (**c**) physical drawing of simulation cabin placed on 50 mm thick Q235 steel target.

**Figure 13 materials-14-06526-f013:**
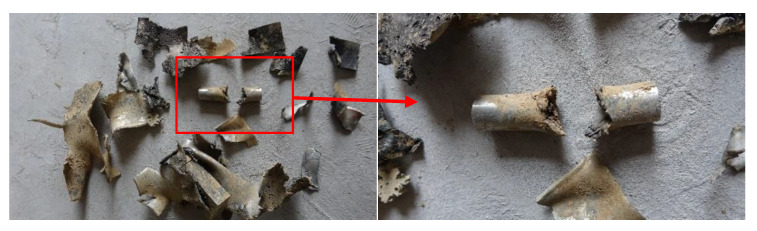
Simulated cabin residual.

**Figure 14 materials-14-06526-f014:**
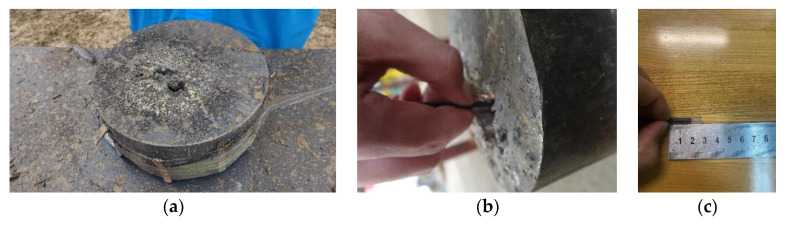
Penetration result of the Q235 steel target with the simulation cabin: (**a**) steel target after penetration, (**b**) jet depth measurement diagram, and (**c**) jet length.

**Table 1 materials-14-06526-t001:** Parameters of each layer slice.

Penetrating Medium		Layer	Name	Materials	Thickness	Distance to The Upper Layer
Simulation Cabin	Control cabin	1	Control cabin shell	Aluminum	2 mm	0
		2	Circuit board 1	Printed circuit board material	2 mm	27.3 mm
		3	Circuit board 2		2 mm	9 mm
		4	Circuit board 3		2 mm	9 mm
		5	Control cabin shell	Aluminum	3 mm	22 mm
	Steering Gear cabin	6	Power connector	Steel	1 mm	100 mm
		7	Analog power 1	Aluminum	0.5 mm	38 mm
		8	Analog power 2		0.5 mm	31 mm
		9	Steering gear cabin shell		5 mm	0.5 mm
		10	Circuit board 1	Printed circuit board material	2 mm	11.5 mm
		11	Circuit board 2		2 mm	9 mm
		12	Circuit board 3		2 mm	9 mm
		13	Steering gear cabin shell	Aluminum	1 mm	16.5 mm
		14	Steering gear shaft 1	Steel	30 mm	5 mm
		15	Steering gear shaft 2		12 mm	7 mm
		16	Electrical machinery		24 mm	17 mm
		17	Circuit board 4	Printed circuit board material	5 mm	25 mm
		18	Power connector bracket	Steel	3 mm	10 mm
	Guidance cabin	19	Guidance cabin shell	Aluminum	4 mm	49 mm
		20	Circuit board	Printed circuit board material	6 mm	9 mm
		21	Analog power 1	Aluminum	0.5 mm	19.5 mm
		22	Analog power 2		0.5 mm	34.5 mm
		23	Analog gyroscope sensor		5 mm	14.5 mm
		24	Guidance cabin shell		3 mm	27 mm
		25	Glass hood	Printed circuit board material	10 mm	28 mm
Steel plate		1	The first steel plate	Q235 Steel	50 mm	2 mm

**Table 2 materials-14-06526-t002:** Material parameters for 8701 explosive.

Material	$\rho \;\left(g/{\mathrm{cm}}^{3}\right)$	D (cm/μs)	$P_{CJ}\;\left(\mathrm{Gpa}\right)$	E (Gpa)	*A* (Gpa)	*B* (Gpa)	$r_{1}$	$r_{2}$	ω	$v_{0}$
Explosive	1.71	0.83	28.6	8.5	524.23	7.678	34	1.1	0.34	1

**Table 3 materials-14-06526-t003:** Material parameters for liner.

Material	$\rho \;\left(g/{\mathrm{cm}}^{3}\right)$	G (GPa)	*A* (MPa)	*B* (MPa)	n	*C*	m	$T_{m}\;(K)$	$T_{room}\;(K)$	$c_{0}$ (cm/μs)	*S*
Copper	8.93	46.5	90	292	0.31	0.025	1.09	1356	293	0.39	1.49

**Table 4 materials-14-06526-t004:** Material parameters for the simulation cabin.

Material	$\rho \;\left(g/{\mathrm{cm}}^{3}\right)$	*E* (GPa)	NUXY	Yield Stress (MPa)
Q235 steel	7.8	207	0.3	355
Aluminum	2.7	71	0.33	90
Printed circuit board material	1.19	7.8	-	-

**Table 5 materials-14-06526-t005:** Material parameter of air.

Material	$\rho \;\left(g/{\mathrm{cm}}^{3}\right)$	γ	$C_{p}$ (kJ/kg·K)	$C_{v}$ (kJ/kg·K)	T (K)	$E_{0}$ $\left(kJ/kg^{-1}\right)$
Air	1.225	1.4	1.005	0.718	288.2	206,800

## Data Availability

The data that support the findings of this study are available from the corresponding author upon reasonable request.

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
