# Peer review of "Study on Penetration Performance of Rear Shaped Charge Warhead"

_materials, 2021, doi:10.3390/ma14216526_

Round 1
Reviewer 1 Report
Paper is interesting and is scientifically sound. There are some minor issues and suggestions given below.
Please explain in more detail conditions on the contact of plates.
Some quick explanations for Plastic_Kinematic and Mat_null models should be given.
Numbering of equations from number 4 is not matched in text.
ALE abrreviation is not explained.
In section 5, subsection and subsubsection titles are missing.
Author Response
Dear reviewer:
Thank you for your letter and the reviewer's review of our title " Study on penetration performance of rear shaped charge warhead " (ID: 1404956). These comments are valuable, very helpful to the modification and improvement of our paper, and also have important guiding significance for our research. We have carefully studied these comments and corrected them, hoping for approval. The revised part is marked in red in the paper. The main corrections in the paper will be replied to you in the form of word file. The file is named "reply to review comments reviewer 1". Please check it.
Finally, thank you very much for your time and energy.
Wish you a happy life!
Thanks again for your reply!
Best Regards.
Ms Yanan Du, Ph.D.

Reviewer 2 Report
This article is very interesting. This kind of warheads show the great potential in the field on the increasing the penetration abilities.
To supplement some lack of information’s I formulate a few remarks:
1) line 45 must be Held - not Hold.
2) line 146 - fomula (4) - please correct to one size the letter ∅,
3) line 401 - Conclusion- there are only presented main results. Please suplement this chapter:
- comparison of the simulation and experimetal results - size of differences (errors) ;
- add some information about the development this problem (project) in the future.
Unfortunately I found only a few references out of China.
Please find below as an example the two fresh articles witch could be associated with the practical application of the issue of your article (could be as a reference of the practical application in introduction section) . Indeed it will supplement contents of the article.
1) Pyka, D., Kurzawa, et all., Numerical and experimental studies of the Łk type shaped charge, Applied Sciences (Switzerland), 2020, 10(19), pp. 1–20, 6742, DOI: 10.3390/app10196742
2)Żochowski, P.; Warchoł, R.; Miszczak, M.; Nita, M.; Pankowski, Z.; Bajkowski, M. Experimental and Numerical Study on the PG-7VM Warhead Performance against High-Hardness Armor Steel. Materials 2021, 14, 3020. https://doi.org/10.3390/ma1411302
Author Response
Dear reviewer:
Thank you for your letter and the reviewer's review of our title " Study on penetration performance of rear shaped charge warhead " (ID: 1404956). These comments are valuable, very helpful to the modification and improvement of our paper, and also have important guiding significance for our research. We have carefully studied these comments and corrected them, hoping for approval. The revised part is marked in red in the paper. The main corrections in the paper will be replied to you in the form of word file. The file is named "reply to review comments reviewer 2". In addition, for the error correction methods mentioned in your modification opinions, I found some literature and compressed it together with "reply reviewer 2 reviewers". Please check it. Please check it.
Finally, thank you very much for your time and energy.
Wish you a happy life!
Thanks again for your reply!
Best Regards.
Ms Yanan Du, Ph.D.

Reviewer 3 Report
The manuscript is interesting, however some minor points can be addressed to enhance and interest the paper:
1-To term To in Equation 3 is not defined
2- The same for the term f in Eq. (4) f is the number
of penetrating the target and ,,,,,. It should be explained.
3-page 6, Term FSI term is not defined
4-Table 2 is not defined and so the type of the used explosive should be mentioned explicitely.
5-Table 4, the steel grade should be defined such as 4340, 4140, ....ETC
6-I think the manuscript address should contain the standoff distance or spaced targets.
Author Response
Dear reviewer:
Thank you for your letter and the reviewer's review of our title " Study on penetration performance of rear shaped charge warhead " (ID: 1404956). These comments are valuable, very helpful to the modification and improvement of our paper, and also have important guiding significance for our research. We have carefully studied these comments and corrected them, hoping for approval. The revised part is marked in red in the paper. The main corrections in the paper will be replied to you in the form of word file. The file is named "reply to review comments reviewer 3". Please check it.
Finally, thank you very much for your time and energy.
Wish you a happy life!
Thanks again for your reply!
Best Regards.
Ms Yanan Du, Ph.D.
